# Peak Bone Mass Formation: Modern View of the Problem

**DOI:** 10.3390/biomedicines11112982

**Published:** 2023-11-06

**Authors:** Karina Akhiiarova, Rita Khusainova, Ildar Minniakhmetov, Natalia Mokrysheva, Anton Tyurin

**Affiliations:** 1Internal Medicine Department, Bashkir State Medical University, 450008 Ufa, Russia; liciadesu@gmail.com; 2Medical Genetics Department, Bashkir State Medical University, 450008 Ufa, Russia; ritakh@mail.ru; 3Endocrinology Research Centre, Dmitriya Ulianova Street, 11, 117036 Moscow, Russia; minniakhmetov.ildar@endocrincentr.ru (I.M.); mokrisheva.natalia@endocrincentr.ru (N.M.)

**Keywords:** peak bone mass, PBM, aBMD, bone mineral density

## Abstract

Peak bone mass is the amount of bone tissue that is formed when a stable skeletal state is achieved at a young age. To date, there are no established peak bone mass standards nor clear data on the age at which peak bone mass occurs. At the same time, the level of peak bone mass at a young age is an important predictor of the onset of primary osteoporosis. The purpose of this review is to analyze the results of studies of levels of peak bone mass in general, the age of its onset, as well as factors influencing its formation. Factors such as hormonal levels, body composition, physical activity, nutrition, heredity, smoking, lifestyle, prenatal predictors, intestinal microbiota, and vitamin and micronutrient status were considered, and a comprehensive scheme of the influence of these factors on the level of peak bone mass was created. Determining the standards and timing of the formation of peak bone mass, and the factors affecting it, will help in the development of measures to prevent its shortage and the consequent prevention of osteoporosis and concomitant diseases.

## 1. Introduction

Peak bone mass (PBM) is the amount of bone tissue that is formed when a stable skeletal state is achieved at a young age [1]. The concept of PBM in a broader sense encompasses peak bone strength. Peak bone strength is characterized not only by mass, but also by density, microarchitecture, microrepair mechanisms, and geometric properties that provide structural strength. Moreover, there is an opinion that the term “peak bone mass” applies only to the individuals, and not to the population.

The peak rate of bone mineral growth occurs at 12.5–33 years in males of European origin [2]. More than 95% of the adult human skeleton is formed by the end of adolescence [3]. In most parts of the skeleton, the total bone mineral mass does not significantly increase from 30 to 50 years, and even several cross-sectional studies have shown that the areal bone mineral density (aBMD) of the proximal femur begins to decrease at the beginning of the third decade [4].

A study of bone metabolism using computer simulations has shown that at a population level, shifts in PBM have a more significant effect on delaying the onset of osteoporosis than delaying the onset of menopause, and the amount of bone mass acquired at the end of the growth period appears to be more important than the rate of its loss [5,6]. An increase in PBM by 10% can delay the onset of osteoporosis by 13 years for most of the population [7], and, consequently, the risk of osteoporotic fractures in the future. Therefore, the analysis of PBM values, its set, and predictors of shortage in order to prevent early osteoporosis seems promising.

The purpose of this review is to analyze the results of studies of PBM levels, as well as the factors involved in its formation.

## 2. Structural Properties of Bone

Bone integrity is maintained by continuous balanced processes of bone resorption and formation; imbalance leads to bone loss, osteopenia, and osteoporosis [8]. Bone strength is partly determined by bone mineral mass and density. The structural properties of bone also contribute to its strength. Characteristics such as thickness, density, and porosity of the cortical bone, as well as trabecular micro architectonics (thickness of trabeculae, number, and distance) together determine the structural strength of the bone. At the same time, the question remains whether the peak of bone strength occurs simultaneously with the peak of bone mass [9].

The mechanical strength of the bone is determined by a number of factors: the size of the bone, the amount of bone tissue in the periosteal membrane, and its spatial distribution, i.e., micro- and macro architectonics, or geometry, as well as the degree of mineralization and structural organization of the organic matrix. During the entire growth period, including puberty, the development of bone mineral mass occurs mainly due to an increase in bone size, that is, the volumetric bone mineral density (vBMD) remains almost constant from birth to the end of the growth period. Similarly, as soon as puberty is reached, sex differences in bone mass arise mainly due to the larger size of bones in men [10]. In girls, bone mineral mass increases more due to endosteal than periosteal accumulation, while in boys, reverse changes were observed with a greater increase in periosteal than endosteal deposition, which leads to an increase in both the outer and inner perimeter of the cortical layer. The thickness of the cortical layer of bone in the vertebral bodies in men is greater than in women. Sexual structural dimorphism is mainly expressed in the frontal axis, which is 10–15% larger in men than in women. These morphological differences and the distribution of the mineral mass of both axial and peripheral bones give the male skeleton greater resistance to mechanical stress [11].

The mechanisms of bone tissue modeling and remodeling are complex and are supported by a balance of bone resorption and bone formation by osteoblasts and osteoclasts. The progenitor of osteoblasts is the mesenchymal stem cell. The ways of its differentiation are controlled by a number of factors. For example, an increase in LRP5/6 (a protein associated with the lipo-77 protein receptor), mediated by canonical Wnt signaling, leads to an increase in bone mass [12]. Osteocytes regulate osteoblast differentiation and anabolic activity, as well as osteoclast activity, through direct contact with dendrites or by secreting important regulatory factors such as sclerostin, an inhibitor of Wnt/β-catenin signaling, and an activator of the cytokine receptor ligand NF-κB (RANKL), the main regulator of osteoclastogenesis [12]. On the other hand, in bones, DKK1 (Dickkopf-related protein 1) and osteocyte-produced sclerostin (SOST) bind the LRP5/6 complex and inhibit Wnt signaling in osteoblasts, limiting osteoblastogenesis and promoting bone resorption. RUNX2 (runt-related transcription factor 2) is the main transcription factor regulating skeletogenesis and is necessary for directing the differentiation of mesenchymal progenitor cells into osteoblasts. The expression of *RUNX2* increases during differentiation of chondrocytes into osteoblasts, but later decreases in mature osteoblasts at the terminal stage [13].

## 3. Peak Bone Mass and Its Absolute Values

In the literature, data about PBM are fragmentary, and as an alternative to the concept of “peak bone mass”, values of bone mineral content (BMC), areal bone mineral density (aBMD—g/cm^2^), and volume bone mineral density (vBMD—g/cm^3^) are used. The gold standard of estimation the accumulation of “bone mass” in numerical terms is aBMD, which accounts for 65–75% of differences in bone strength [14]. aBMD is determined by dual-energy X-ray absorptiometry and is calculated by dividing the amount of bone minerals (bone mineral content (BMC in grams)) by the area scanned bone (cm^2^) [15]. vBMD is also used, which is defined as BMC/V (bone volume) and the V value is calculated based on the projection area of the bone [16]. At the population level, the PBM is reached when age-related changes in bone metabolism cease to be positive and reach a maximum value. However, PBM is regarded as one of the most important predictors of bone strength. There are few studies describing the significance of aBMD in young adults. In addition, the age of reaching the peak of bone mass and the absolute values of aBMD differ in different sources, because the comparison groups are heterogeneous, namely, different age groups, sex, and different examination methods (systems, skeletal areas) are described. This creates a problem in interpreting the values of aBMD in young people, despite the presence of the Z-criterion. The main literature references of aBMD in young adults are presented in Appendix A. Appendix A presents the results obtained by X-ray absorptiometry with determination of aBMD for the lumbar spine and/or femoral neck for young people aged 11 to 43.19 years. Two small studies in Russia measured aBMD in young adults. Zakharov I.S. et al. (2014) described aBMD of the lumbar spine among women aged 16 to 80 years and older, in which the highest aBMD was achieved at the age of 20–29 years (1.219 g/cm^2^) [17]. Paskova et al. also analyzed aBMD of the lumbar spine, but for men aged 20 to 78 years, while the PBM was at the age of 21–25 years (1.20 ± 0.15 g/cm^2^) [18]. Anne Looker et al. (1997) summarized the results of a large population-based study NHANES III (USA). They described the achievement of PBM for both men and women, according to dual-energy absorptiometry of the femoral neck, at the age of 20–29 years. They formed reference values from these indicators values for assessing osteopenia in the older age group. The maximum value of aBMD for women was 0.86 ± 0.12 g/cm^2^ (N = 382), and for men, it was 0.93 ± 0.137 g/cm^2^ (N = 409), which is slightly less than the values obtained in the population of the Russian Federation [19]. In a small UK study, Henry et al. (2004) evaluated aBMD, volumetric mineral density, and BMC. The subjects were divided into two groups: 11–19 and 20–50 years old, respectively. The PBM of the lumbar spine was achieved for women at the age of 29 years and amounted to 0.970 g/cm^2^ (N = 69), and for men at the age of 22 years—1.05 g/cm^2^ (N = 63), which, in general, correlates with the results of NHANES III [20]. Three studies in Sweden analyzed aBMD among young men. Karlsson (2001) et al. evaluated the effect of sports on aBMD in young men, healthy volunteers were examined as controls, whose PBM was at the age of 24.4 ± 0.6 (N = 24) and lambar spine aBMD was LS 1.27 ± 0.03 g/cm^2^ [21]. Further Lorentzon et al. (2005), according to the results of the Gothenburg Study of Determinants of Osteoporosis and Obesity (GOOD), determined that among young men aged 18.9 ± 0.6 year, the average aBMD for L2-L4 was 1.24 ± 0.15 g/cm^2^ [22], which correlates with the results of Karlsson, but the age of PBM was not reached. Tveit et al. (2014) described in a study among men aged 18–85 years the effect of such a sport as football on bone tissue. The highest values of aBMD were achieved at the age of 23.96 ± 3.85 years (N = 135) and amounted to 1.36 g/cm^2^ [23]. Bakker et al. (2003) analyzed lifestyle, physical activity, and aBMD L2-L4 in men and women in three age groups: 27, 32, and 36 years, with the maximum values of aBMD occurring at 27 years (N = 83): 1.170 ± 0.159 g/cm^2^ [24]. Two studies in Australia have evaluated aBMD of the lumbar spine for men and women. Liberato (2015) analyzed the influence of lifestyle, nutrition, and physical activity on aBMD, the mean age was 21.8 ± 2.18 years and limbar spine aBMD was (1.16 ± 0.15 g/cm^2^) [25]. For women, as found by Jones et al. (2000), at the age of 33.5 years, the values were 1.07 ± 0.12 g/cm^2^ [26]. Another study in Finland analyzed bone mineral density at the femoral neck in young men and women with and without a history of fractures. aBMD of the control group for men (26 ± 4 years) was 0.952 ± 0.127 g/cm^2^, and 0.900 ± 0.132 g/cm^2^ for women (22 ± 5 years), respectively [27].

In China, Ho et al. (1993) analyzed bone mineral density in women aged 21 to 40 years. The highest values of aBMD were described for the age of 29–32 years (N = 57) for the lumbar spine (1.03 ± 0.12 g/cm^2^) and for the femoral neck (0.86 ± 0.11 g/cm^2^) [28], which correlates with the data obtained in the Netherlands. Yanping Du et al. (2018) also evaluated aBMD of lumbar spine aBMD. For men and women, the highest values of aBMD were achieved at the age of 21–30 years and amounted to 0.986 ± 0.150 g/cm^2^ (N = 113) for men and 0.970 ± 0.100 g/cm^2^ (N = 128) for women, respectively [29]. In Japan, a large study by Orito S. et al. (2009) evaluated various clinical and instrumental characteristics of the musculoskeletal system, where aBMD L2-L4 in women was 0.961 ± 0.119 g/cm^2^ for the age group of 12–30 years (N = 1322) [30]. In general, lumbar spine aBMD scores in young adults in the Russian population are broadly consistent with results for similar parts of the skeleton in Swedish and Dutch populations in comparable age and gender groups, and are somewhat superior to those in Asian, UK, and Australian populations.

Studies of bone mineral density of the femur neck among young subjects are less common. However, absolute values of aBMD are higher in the Swedish population compared to the Asian population, but further studies with larger numbers of participants are required. The highest values of aBMD are achieved by Caucasians at the age of 20–29 years, and Asians 29–32 years, respectively. Thus, PBM ranged from 0.79 ± 0.12 to 1.18 ± 0.03 g/cm^2^ in women and from 0.93 ± 0.137 to 1.27 (0.14) g/cm^2^ in men at the femoral neck.

## 4. Factors Affecting PBM

According to the literature data, there are a number of factors affecting both the set of PBM and bone formation in general [31]. The main factors and possible pathogenetic mechanisms of their influence on bone metabolism are presented in Figure 1 and Figure 2.

### 4.1. Hormonal Background

One of the key roles in the regulation of bone metabolism is played by sex hormones. Thus, estrogens limit the production of osteoclastogenic cytokines produced by osteoblast cells and thus inhibit osteoclast formation and bone resorption [32]. Changes in the hormonal background in different age periods have a significant effect on bone metabolism. The effect of hormonal background was studied in mice in the most detail. For example, deletion of the androgen receptor (AR) in male mice causes high bone turnover, increased bone resorption, and reduced cortical and cancellous bone mass. Targeted deletion of AR in mature osteoblasts, however, reduces the mass of the bone spongy substance, but does not affect the cortical layer of bone [33]. Over the past 10 years, extensive data have indicated the involvement of oxidative stress in increased bone resorption associated with estrogen or androgen deficiency [34]. RANKL and MCS-F (macrophage colony-stimulating factor) are two cytokines that are necessary for the formation of osteoclasts [35]; they stimulate the intracellular accumulation of H_2_O_2_, which is necessary for osteoclast adaptation, differentiation, and survival [36]. This hypothesis has been proven empirically: in mice capable of synthesizing human catalase, an enzyme that utilizes H_2_O_2_ in mitochondria, there was an increase in cortical and spongy bone mass due to a decrease in the number of osteoclasts [33]. In adulthood, certain endocrine pathologies associated with estrogen, such as complete androgen insensitivity syndrome due to complete androgen resistance, premature ovarian failure, or Turner syndrome, can lead to a decrease in aBMD in women at the level of the lumbar spine and femoral neck [37].

On the other hand, it is interesting that gender-supportive hormone therapy in trans women (with estradiol and antiandrogens) and in trans men (with testosterone) resulted in an increase in bone metabolism in young trans men, while it decreased in trans women, demonstrating the crucial role of estrogen in the regulation of bone resorption [38].

### 4.2. Microbiome

Intestinal microbiota makes a certain contribution to the metabolism of connective tissue and bone tissue. Yan et al. (2017) reported that short-chain fatty acids (SCFAs) produced by the microbiota induced insulin-like growth factor 1 (IGF-1), which promotes bone growth [39]. Treatment of mice with microbiota metabolism products, including SCFAs, propionate, and butyrate, significantly increases bone mass and prevents postmenopausal and inflammation-induced bone loss, and propionate and butyrate induce metabolic reprogramming of osteoclasts, which leads to suppression of bone resorption activity by osteoclasts [40,41,42]. The consumption of indigestible food components is one of the ways in which it is possible to change the gut microbiota to improve the health of the host. Dietary components, such as prebiotic dietary fiber, are associated with positive shifts in the composition of the intestinal microbial community [43]. A study conducted among adolescents showed that the consumption of various prebiotics, such as galactooligosaccharides and soluble corn fiber, which can be fermented to SCFAs, led to increased absorption of calcium in the intestine and was associated with the relative content of *Parabacteroides*, *Bifidobacterium*, *Bacteroides*, *Butyricicoccus*, *Oscillibacter*, and *Dialister* species measured in feces, and the change in the intestinal microbiome towards enrichment with bacteria of the genera *Dialister* and *Faecalibacterium*, on the contrary, was associated with the presence of osteoporosis [44]. In addition, in clinical trials, it was reported that the consumption of SCFAs in women could improve bone metabolism with increased activity of bone-specific alkaline phosphatase [45]. Supplements with *Bacillus subtilis*, Lactobacilli, and multi-species probiotics have demonstrated a beneficial effect not only on the human gut microbiota [46], but also on markers of bone metabolism [47]. Strain-specific probiotics can reduce oxidative stress by producing several antioxidant molecules, e.g., glutathione, folic acid, and exopolysaccharide. In addition, short-chain fatty acids produced by some gut microbiota may also help reduce oxidative stress by promoting the production of antioxidant molecules [48].

### 4.3. Quality Body Composition

The understanding of the effect of adipose tissue on bone metabolism is still ambiguous, and often the results of studies contradict each other. It is worth considering the adipose tissue of the bone marrow and adipose tissue in general. Mesenchymal stromal cells (MSCs) are multipotent progenitors capable of differentiation into osteoblasts and can potentially serve as a source of cell therapy for bone regeneration. Many factors have been shown to regulate the differentiation of MSCs into the osteogenic lineage, such as the cyclooxygenase-2 (COX2)/prostaglandin E_2_ (PGE_2_) signaling pathway, which is critical for bone repair. PGE_2_ binds four different EP1-4 receptors (prostaglandin receptors) [49]. Thus, adipocytes themselves stimulate the differentiation of MSCs into adipocytes, and not into osteoblasts. In addition, adipocytes in the bone marrow microenvironment release a range of pro-inflammatory and immunomodulatory molecules that enhance osteoclast formation and activation, thereby contributing to bone fragility. In vivo analysis of the relative content of saturated and unsaturated fatty acids in the bone marrow indicates a locally dependent marrow fat composition and an association between an elevated unsaturation index and bone health. Most diseases with bone loss, in which an altered composition of the bone marrow develops, have aging and/or chronic inflammation as common factors. Both saturated and unsaturated fatty acids form lipid forms that are active mediators of the inflammatory process [50]. However, the intervention of alpha-lipoic acid (ALA) inhibited the RANKL-induced proliferation and differentiation of osteoclasts. ALA also inhibited bone resorption activity, suppressed RANKL-induced transcription factors c-Fos, c-Jun, and NFATC1 (nuclear factor of activated T cells) in combination with markers of osteoclasts such as TRAP (prediction of affinity for transcription factor), OSCAR (receptor associated with osteoclasts), cathepsin K, and β3-integrin [51]. On the other hand, excessive accumulation of adipose tissue performs the role of a mechanical vertical load contributing to the build-up of bone tissue. In this case, bone quality and structure are the result of a balance of inflammatory and mechanical incentive stimuli [52].

Data on the influence of sex on body mass index (BMI) and aBMD are also contradictory. There is a point of view that aBMD values increase with an increase in BMI only in men [53]. Nevertheless, a number of authors have noted that increasing BMI to the values of the “metabolic syndrome” leads to decreasing aBMD for both men and women [54]. In another study among young Hispanic and non-Hispanic girls, Megan Hetherington-Rauth et al. found a positive contribution of fat mass to bone strength in vBMD; however, negative associations were also found with the content and thickness of the cortical bone of the radius [55]. Also, Zeyu Xiao et al. in a sample of young Chinese adults found that both total lean body mass and fat mass were significantly positively associated with aBMD in both genders [56]. Nielsen et al. obtained results that individuals aged 15–19 years who lost weight during follow-up showed slower progression of aBMD gain compared to those who gained weight, but weight loss or BMI reduction over 2 years was not associated with a net loss of aBMD [57].

In a study on FAT-ATTAC transgenic mice, Lagerquist et al. induced adipocyte apoptosis and assessed bone metabolism by X-ray absorptiometry and found no effect of adipose tissue reduction on bone resorption [58].

Bones and muscles are two deeply interconnected organs with integrated growth and locomotion [59]. In a study involving 416 women and 334 men aged 20 to 30 years in Vietnam, the association of body composition and aBMD at the lumbar spine and femoral neck was assessed using dual-energy X-ray absorptiometry. Peak aBMD in men was higher than in women, and the difference was more pronounced in the femoral neck region than in the lumbar spine, and fat-free mass was the only predictor of aBMD for both men and women. Each kilogram of increase in muscle mass was associated with an increase in BMD by about 0.01 g/cm^2^ [60]. Winther et al. also found that higher levels of aBMD corresponded to higher levels of fat-free mass in both genders, but higher aBMD at higher levels of fat mass was found only in girls [61].

Muscle paralysis also contributes to bone mineral density. Paralysis caused by botulinum toxin causes bone loss in adult mice and slows down the healing of fractures [62].

Bones and muscles act as secretory endocrine organs that affect each other’s function. Biochemical crosstalk occurs through myokines such as myostatin, irisin, interleukin IL-6, IL-7, IL-15, insulin-like growth factor-1, fibroblast growth factor, and β-aminoisobutyric acid, as well as through factors of bone origin, including FGF23, prostaglandin E_2_, transforming factor growth of β, osteocalcin, and sclerostin [63]. Myostatin is member of the transforming growth factors-beta (TGF-beta) superfamily, which is highly expressed in skeletal muscles. Loss of myostatin function in mice led to an increase in muscle mass, increased bone formation, and an increase in bone cross-section in most anatomical areas, including limbs, spine, and jaw in mice [64]. Myostatin inhibitors such as ACVR2B/Fc, a soluble myostatin decoy receptor, have been shown to prevent loss of both muscle and bone tissue in models of muscular dystrophy [65]. In vitro studies on mice have also shown that myostatin enhances RANKL-induced osteoclastogenesis [66]. Myostatin is also expressed in the early stages of fracture healing, and myostatin deficiency leads to an increase in the size and strength of the callus during fracture. Taken together, these data suggest that myostatin may affect the proliferation and differentiation of osteogenesis progenitor cells and that myostatin antagonists and inhibitors may be therapeutically useful for increasing both muscle mass and bone. Myostatin directly affects osteocyte function by suppressing exosomal *miR-218* microRNA and thus inhibits osteoblast differentiation [67].

### 4.4. Smoking

The negative impact of smoking on various systems and organs is multifaceted. Mattias Callréus et al. investigated the effects of smoking among women aged 25 years, but statistically significant differences were found only in the levels of aBMD of the femoral neck. Similarly, lower aBMD, in comparison with the control, persisted for up to 24 months after quitting smoking, becoming comparable to those who had never smoked after 24 months [68]. To further evaluate these outcomes, a meta-analysis of 14 prospective studies, information was conducted, which showed that, compared with those who have never smoked, cigarette smoking increased the risk of hip fracture in men, especially in current smokers [69]. Another study has also found an association between smoking and an increased risk of fractures [70]. Both active and passive smoking negatively affect bone mass; quitting smoking seems to reverse the effect of smoking and improve bone health [71]. In men, regardless of age, method, and site of bone density measurement, cross-sectional studies showed that smokers had significantly lower aBMD than non-smokers [72]. In bone studies, smoking was associated with lower aBMD, increased risk of fractures, periodontitis, loss of alveolar bone, and rejection of dental implants [73]. In a large NHANES III study involving 14,500 subjects, the bone mineral density of the femoral neck in smokers was numerically lower than in never-smokers, but the statistical significance of the difference was not reached [74,75]. The data on cannabis smoking are contradictory. Thus, bone mineral density (Z-criterion) was significantly lower in avid cannabis users compared to the control group in the lumbar spine, hip neck, and hip as a whole [76]; however, it was found that the cannabinoid receptors CB1 and CB2 were expressed in bones and regulate bone homeostasis in rodents and humans. Cannabis treatment has been shown to improve fracture healing in rats [77].

### 4.5. Nutritional Deficiencies (Calcium, Vitamin D, Phosphorus), Lipids, and Food Character

The contribution of calcium and vitamin D to the quality of bone tissue is beyond doubt; the use of sufficient amounts of calcium and vitamin D in childhood and adolescence is of the greatest importance. Vitamin D synthesized in the skin or absorbed with food undergoes a multi-step enzymatic transformation into its active form, 1,25-dihydroxyvitamin D, [1,25(OH) 2 D], followed by interaction with the vitamin D receptor (VDR) to modulate the expression of the target gene [78]. Vitamin D can support skeletal health and improve bone mineralization by increasing calcium absorption in the intestine, reducing secondary hyperparathyroidism, and reducing bone resorption [79]; moreover, increased calcium intake in combination with vitamin D reduces the rate of loss of minerals in the bones without harm to the intestinal microbiota. Guo-Hau Gou et al. described the positive effect of daily intake of vitamin D and magnesium on hip neck aBMD in female participants aged 8–11 years [80]. Zhou also proved that higher intake of calcium and vitamin D was associated with higher total hip and hip neck aBMD in young men (16–18 years old), and cumulatively high levels of calcium and vitamin D intake over time contributed to better maintenance of aBMD in the lumbar spine and femurs in adult women [81]. Neville et al. also described the positive effect of vitamin D on hip neck aBMD in adolescent girls, but not in the lumbar spine [82]. Krstic et al. described the positive effect of vitamin D intake and physical activity on young mice, explaining the mechanism by hypomethylation of the RXRA (Retinoid X Receptor Alpha) gene DNA, which together with the vitamin D receptor forms a mechanism that transmits the nuclear effects of vitamin D [83].

Hui Li et al. obtained results according to which vitamin C intake with food has a threshold effect on the bone mass increase in male adolescents, and vitamin E intake with food can be a positive predictor of bone mass increase in adolescent girls; however, the aBMD of the calcaneus was evaluated by ultrasound [84].

Elham Z. Movassagh et al. described a vegetarian diet rich in dark green vegetables, eggs, whole grains, 100% fruit juice, legumes, nuts, seeds, added fats, fruits, and nonfat milk during adolescence as being positively associated with whole body BMC [85].

The question of the effect of dietary lipids on bone tissue remains open. In a mouse study, a high-lipid diet (HLD) resulted in osteoclast activation, a decrease in trabecular bone volume, along with an increase in bone marrow deposition compared to a low-fat diet group [86]. In addition, HLD led to an increase in bone marrow adipose tissue and changes in the bone marrow microenvironment along with a pro-inflammatory environment, which could contribute to a negative effect on bone metabolism. At present, the effect of adipose tissue on bone metabolism is not fully understood [87].

### 4.6. Physical Activity, Lifestyle, Psychoemotional Status

There is a positive relationship between the incidence of fractures and the level of physical activity due to the increased risk of falls during physical activity. Thus, although physical activity is critical for bone modeling, children with higher levels of physical activity are more likely to have fractures [88]. Weekly physical activity has a positive effect on BMC and bone mineral density in both boys and girls during puberty when exercising 1–2 times a week [89]. Physical exercises at school in terms of time are effective for increasing aBMD and/or vBMD in children and adolescents, but should include high-intensity exercises, such as high-impact jumps [90]. Physical activity 3 times a week for 40 min in ball games or circular strength training throughout the school year improves bone mineralization and some aspects of muscle fitness of children aged 8–10 years, suggesting that well-organized intensive physical education classes can positively affect the development of the musculoskeletal system and health in children early age [91].

Sleep also affects bone mineralization. An unbalanced sleep pattern contributes to bone loss and an increased risk of osteoporosis. Thus, healthy sleep contributes to the prevention of osteoporosis [92].

Psychoemotional status, as well as associated pathological conditions, also affect bone health. So, depression was associated with lower aBMD, especially in the spine, in white men, and non-highly educated populations. Moreover, people with depression were more likely to suffer fractures and osteoporosis [93].

### 4.7. Early Antropometric Characteristics

Body length at birth and height under the age of 7 years were positively associated with mineral density of the femoral neck, and growth in all studied age periods was positively associated with the area of the spine. Growth under the age of 7 years was associated with the mineral density of the femoral neck [94]. Bone mineral density may also be affected by birth weight, among other things. In a meta-analysis by Baird et al. (14 studies involving men and women aged 18–80 years), a positive relationship was found between birth weight and aBMD of the lumbar spine in adulthood, and this relationship was stronger for women [95]. This hypothesis was also confirmed in the PEAK-25 study conducted in Sweden (1061 women aged 25 years) [96].

### 4.8. Heredity and Genetics

Data regarding the association of PBM with genes and polymorphic variants in humans are incomplete [97,98]. Wei-jia Yu and colleagues (2020) reported associations between polymorphic variants of the gene *LGR4* (Leucine Rich Repeat Containing G Protein-Coupled Receptor 4) and bone metabolism. Their study involved 1296 participants from nuclear families (mother, father, son) and they found an association of polymorphic variants *rs11029986* and total hip aBMD and *rs12796247*, *rs2219783* with lumbar spine aBMD [99]. Another study by Zheng et al. (2016) described associations of six SNPs (*rs6126098, rs6091103*, *rs238303*, *rs6067647*, *rs8126174*, and *rs4811144)* in the *CTNNBL1* gene (Beta-catenin-like protein 1) and peak bone mineral density of the lumbar spine, femoral neck, or the entire femur [100]. Zhao et al. (2017) conducted a study on 1296 Chinese men and found positive associations between *rs9585961 METTL21C* (Methyltransferase Like Protein 21C) and aBMD of the lumbar spine and femoral neck, as well as *rs9518810* and aBMD of the femoral neck [101]. In a separate study, He et al. (2011) analyzed 401 Chinese nuclear families and described an intrafamily negative relationship between allele C *rs16878759* and PBM of the lumbar spine. They also found that the CCC haplotype (containing *rs12699800*, *rs16878759*, and *rs17619769*) had a significant intrafamily association with PBM of the lumbar spine [102]. Chesi et al. (2019) conducted a study using a donor culture of primary MSC. *ING3* (Inhibitor of Growth Family Member 3) and *EPDR1* (Ependymin Related 1) knockdowns disrupted osteoblast differentiation and increased MSC adipogenic differentiation. *ING3* knockdown increased adipogenesis by 8 times, and *EPDR1* knockdown by 3.5 times [103]. Furthermore, this group of authors, in a repeated experiment with editing the *EPDR1* gene by CRISPR-Cas 9 on an immortalized MSC hFOB1.19 cell culture, confirmed the important role of *EPDR1* in osteoblast differentiation [104].

The search for genes involved in bone metabolism continues in mouse models. In modified mice with Wnt16^−/−^ knockout, the thickness of the cortical layer of bone and bone strength were reduced [105]. An intronic variant *rs2566752* was associated with aBMD of the spine. A less common allele C from this locus was associated with increased spinal aBMD. This allele was also found to be associated with a reduced risk of fracture in the TwinsUK cohort [106]. Fibroblast growth factor receptor 1 (*FGFR1*) is an important molecule for skeletal development and bone remodeling. Mice lacking *FGFR1* in osteocytes showed an increase in trabecular bone mass at 2 and 6 months of age as a result of increased bone formation and reduced resorption [107].

## 5. Conclusions

This article discusses the formation of PBM and the factors that can affect it. PBM is reached at the age of 20–30 years in different populations, with bone mineral density values ranging from 0.79 ± 0.12 to 1.18 ± 0.03 g/cm^2^ in women and from 0.93 ± 0.137 to 1.27 (0.14) g/cm^2^ in men in the femoral neck. A generalized scheme for the formation of PBM is shown in Figure 2. The formation and resorption of bone tissue are continuous processes of bone metabolism. However, at a young age, before reaching PBM, accumulation prevails over resorption and determines the pool of bone tissue. The lower PBM leads to the early development of osteoporosis, disability, and a decrease in the duration and quality of life. In this regard, it is critically important to understand the main mechanisms and factors influencing the formation of PBM in order to develop methods to increase it, which can serve as a significant means of preventing osteoporosis and concomitant diseases.

## Figures and Tables

**Figure 1 biomedicines-11-02982-f001:**
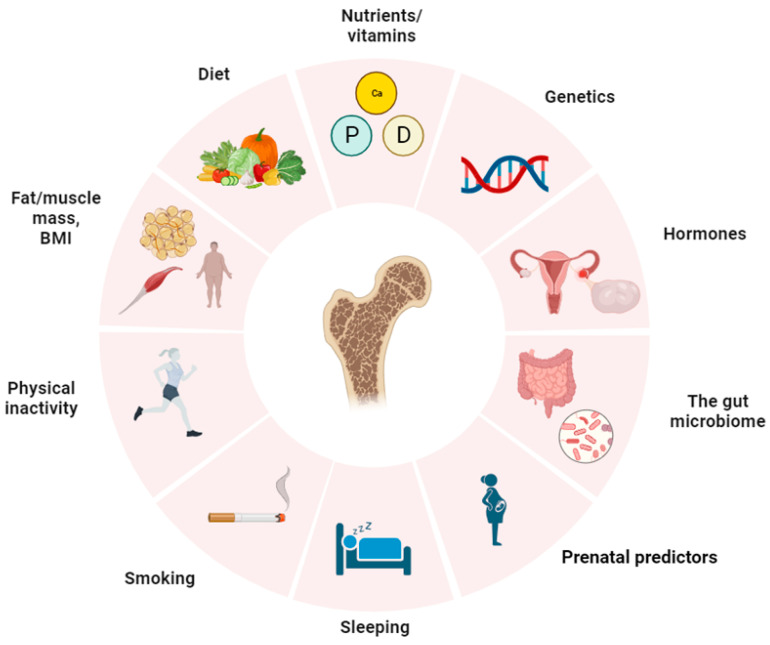
Endogenous and exogenous factors affecting PBM.

**Figure 2 biomedicines-11-02982-f002:**
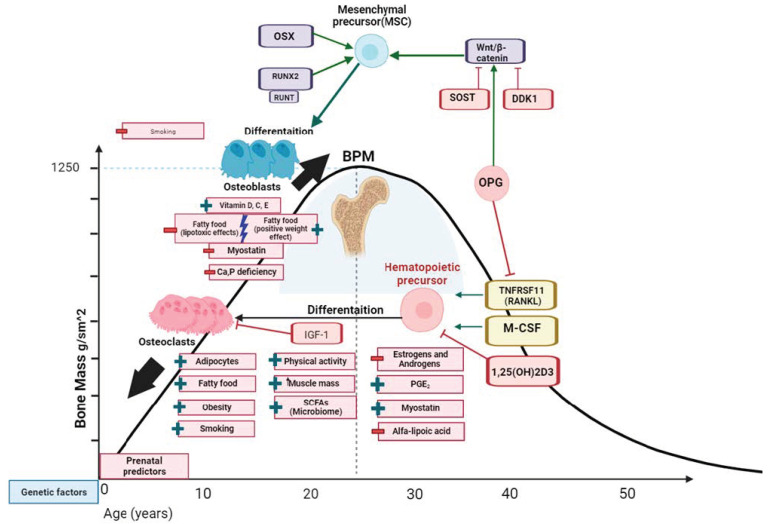
Mechanisms of the influence of risk factors on the formation of peak bone mass. The figure shows a simplified version the molecular mechanisms of modeling and remodeling of bone tissue and possible factors of an endogenous and exogenous nature that can influence certain links of pathogenesis. MCSF—macrophage colony-stimulating factor; TNFRSF11 (RANKL)—tumor necrosis factor superfamily member 11 (tumor necrosis factor ligand superfamily member 1); PGE_2_—prostaglandin E_2_; SCFAs—short-chain fatty acids; OPG—osteoprotegerin; SOST—sclerostin gene; DDK1- Dickkopf WNT signaling pathway inhibitor 1; OSX1—osteoblast-specific transcription factor Osterix; RUNX2—runt-related transcription factor 2; + denotes positive influence; − negative influence.

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
