# Peer review of "Peak Bone Mass Formation: Modern View of the Problem"

_biomedicines, 2023, doi:10.3390/biomedicines11112982_

Round 1

Reviewer 1 Report

Comments and Suggestions for Authors

The manuscript by Akhiirarova and collaborators reviews varied data in regard to peak bone mass. Although the manuscript includes valuable information, the data is generally presented in a disorganized non-coherent nmanner, both in paragraph structure and flow. It also requires a thorough review of English.

Examples of this:

-        In many of the subsections in section 3, the authors begin with studies and data on how different factors affect BMD in general, while clinical data on peak bone mass is sometimes buried in the section (e.g. section 3.3), instead of being the focus. Considering that this is a review on peak bone mass, which is life-stage specific, in my opinion, more weight should be put on the clinical data available on how these factors have shown to affect peak bone mass specifically, and on preclinical studies and/or clinical studies in different age populations, that address how these factors may affect BMD.

-        The content in the second paragraph of the introduction is difficult to understand. Please review

-        In the sentence “However, there is also a tendency for aBMD to be more prevalent in the Swedish population compared to the Asian population, but more participants are required”. What do the authors mean with aBMD to be “more prevalent”? Do you mean that is more frequently analyzed and reported? This sentence is overall not well written.

-        The sentence “According to the literature data, there are a number of factors affecting both the set of PBM and bone formation in general. The main factors and possible pathogenetic mechanisms of their influence on bone metabolism are presented in Figures 1 and 2.”  should be place under section 3.

-        The author state “The presence of a hormonal background in different age periods has a significant impact on bone metabolism.” but the following sentences do not address this statement. Please be more coherent with the sentence flow.

-        The sentence “Thus, deletion of the androgen receptor (AR) in male mice causes high bone metabolism, increased bone resorption, and reduced cortical and cancellous bone mass.” Frist, eliminate thus (this adverb is not correctly used here) and change it for “for example”. There are other instances in which “thus” is used incorrectly throughout the manuscript. Also, it would be more correct in this sentence to employ “high bone turnover” instead of “bone metabolism”. 

-        The authors use the terms “to confirm this hypothesis” after showing data from studies. The term hypothesis is therefore not suitable. In my opinion it would be more adequate to use something like: “to further evaluate these outcomes”.

-        Psychoemotional status should not be under the same section a s “physical activity or lifestyle”. It’s a different aspect.

-         

-         

Other comments:

-        Given that Biomedicine is not a specifically journal within the “bone field”, it would be of interest to define and describe many of the specific “bone-field”terms employed through the paper (e.g. bone modeling and remodeling, BMC, differences between vBMD and aBMD etc).

-        Better use “sex” than “gender”.

-        In regard to the sentences: “The precursor of osteoblasts is the mesenchymal stem cell. An increase in LRP5/6 (lipo-77 protein receptor-related protein) mediated canonical Wnt signaling leads to an increase in bone mass.” These 2 sentences don’t flow. Also, this paragraph describes signaling pathways related to bone remodeling. I’m not sure it fits properly under the section “Structural Properties of bone”.

-        Sclerostin is introduced earlier in the paragraph and is mentioned after with an acronym. Please define the acronym the first time sclerostin is mentioned. Also, in defining acronyms, first the whole word should be written followed by the acronym employed in parenthesis, not the other way around.

-        In the regard to DC1(Dickkopf-related protein 1): the gene/protein symbol for Dickkopf-related protein 1 is DKK1. I’ve checked Gene cards and haven’t found DC1 as an alternative. Also, the authors could also mention that DKK1 is produced by osteoblasts as it is stated that SOST is produced by osteoclasts.

-        Perhaps, if the main literature results of mean aBMD in young populations is already presented in Apendix 1, the authors could just do a discussion of the main outcomes instead of listing them in the form of a paragraph.

-        I think that the section 3.7. Prenatal predictors should be renamed as the authors also include data on childhood growth (which is not prenatal). Perhaps something like “early antropometric characteristics” would fit better?

-        In the section “Heredity and genetics” I think is enough to include the genes and say that polymorphism and/or variants were found” and include the reference to the study, instead of inserting the specific rs number.

-        The meaning of the positive and negative signs  in Figure 2 should be further described in the legend.

Comments on the Quality of English Language

Comments on this aspects are mentioned in the review addressed to the authors.

Reviewer 2 Report

Comments and Suggestions for Authors

This review article summarized recent progress on the factors affecting PBM. Overall the review is well written with two summarized figures and 1 table.  There is following review article which reviewed similar aspects of factors should be mentioned. (Front Med . 2021 Feb;15(1):53-69. doi: 10.1007/s11684-020-0748-y. Epub 2020 Jun 9. Factors influencing peak bone mass gain. Xiaowei Zhu, Houfeng Zheng. PMID: 32519297 DOI: 10.1007/s11684-020-0748-y). Please mention or cite.

Following are some of my comments:

Line 56, distance should be separation?  

Line 196, bone is not a connective tissue.

Line 229 extra period.

Line 247, two words with no space in between.

Lines 241-244, the authors got opposite on what are transcription factor and what are markers of osteoclasts. Clearly, c-jun, c-fos and NFATC1 are transcription factors and TRAP, cathepsin K and β3-integrin are markers of osteoclasts.

Line 245, “Performs the role of load”, I do not understand what the authors wanted to say.

Lines 265, extra: “.”.

When authors say association of factors (such as SNP) with BMD, BMC or any other parameters, they should state positive association or negative association, otherwise readers have to go to original literature.  

Line 425-427, “A shortage of PBM leads to the formation of early osteoporosis”.   May be better “The lower PBM leads to the early development of osteoporosis”.

Fig. 2 Schematic summary, please state what are the (+) and (-) mean for each factor. It is not straightforward to elaborate what factors play what role for the PBM. Maybe up and down arrows will be better. 

Comments on the Quality of English Language

Overall English writing is good with minor edit and proofread needed. 

Reviewer 3 Report

Comments and Suggestions for Authors

The purpose of this review is to analyze the results of studies of PBM levels, as well  as the factors involved in its formation. This article discusses the formation of PBM and the factors that can affect it.

Introduction. The authors make a brief description of the problem and establish the hypothesis. The authors carry out a novel non-narrative review. They provide information on the factors associated with PBM 

Author Response

Dear reviewer, the team of authors thanks you for valuable and very detailed comments.

Round 2

Reviewer 1 Report

Comments and Suggestions for Authors

The authors have addressed most of my comments and I consider that the manuscript has improved substantially. While I personally think that the manuscript's structure would benefit by some changes I suggested in my initial review, and that the authors prefer not to undertake, I consider that overall, the manuscript is acceptable for publication.

Comments on the Quality of English Language

See comments above.